# Krill faecal pellets drive hidden pulses of particulate organic carbon in the marginal ice zone

A. Belcher [1], S.A. Henson[2], C. Manno[1], S.L. Hill [1], A. Atkinson [3], S.E. Thorpe [1], P. Fretwell[1], L. Ireland[1] & G.A. Tarling[1]

The biological carbon pump drives a flux of particulate organic carbon (POC) through the ocean and affects atmospheric levels of carbon dioxide. Short term, episodic flux events are hard to capture with current observational techniques and may thus be underrepresented in POC flux estimates. We model the potential hidden flux of POC originating from Antarctic krill, whose swarming behaviour could result in a major conduit of carbon to depth through their rapid exploitation of phytoplankton blooms and bulk egestion of rapidly sinking faecal pellets (FPs). Our model results suggest a seasonal krill FP export flux of 0.039 GT C across the Southern Ocean marginal ice zone, corresponding to 17–61% (mean 35%) of current satellite-derived export estimates for this zone. The magnitude of our conservatively estimated flux highlights the important role of large, swarming macrozooplankton in POC export and, the need to incorporate such processes more mechanistically to improve model projections.

[1] British Antarctic Survey, Cambridge CB3 0ET, UK. [2] National Oceanography Centre, Southampton SO14 3ZH, UK. [3] Plymouth Marine Laboratory, Prospect Place, The Hoe, Plymouth PL1 3DH, UK. Correspondence and requests for materials should be addressed to A.B. (email: annbel@bas.ac.uk)

The flux of particulate organic carbon (POC) through the ocean via the biological carbon pump (BCP) is tightly coupled to atmospheric levels of carbon dioxide ($CO_2$)[1]. However, limited observations result in poorly constrained models of POC flux, particularly in the Southern Ocean. The rate at which the flux of sinking POC declines with depth can be quantified and modelled by the attenuation rate. However, studies of POC flux attenuation struggle to capture episodic, intense, or localised export events because of limited sampling and the patchy nature of such events in time and space. Hence these events may be under-represented in flux estimates. These intense events may be particularly efficient in transferring POC to the ocean's interior as they may be driven by high primary production, and subsequent faecal pellet production by consumers, which overwhelms the grazing capacity of the community of detrital feeders[2–4]. In this study, we focus on the POC flux generated by the faecal pellets of Antarctic krill.

Antarctic krill (*Euphausia superba*) comprise the highest individual species biomass of any metazoan in the Southern Ocean; swarms can extend over areas of ~100 km2[5]. Krill are thus an important part of Southern Ocean ecosystems, forming a key link between primary producers and higher trophic levels[6]. Being one of the largest epipelagic crustaceans, krill also produce large faecal pellets (FPs), which sink at speeds of hundreds of metres per day[2, 4], making them important agents in carbon export. In this way, carbon that originated in the atmosphere is transferred to the deep sea where it can be sequestered for periods of centuries or more[7]. Krill FPs have been found in high numbers in sediment traps in both the upper and deep ocean[8–10]. Although some carbon from krill FPs may be retained in the upper mixed layer through coprorhexy, coprophagy, and remineralisation[11, 12], two combining factors, namely the rapid sinking speeds of krill FPs and the occurrence of krill in large swarms, mean that atypically large and episodic krill FP fluxes can occur in certain regions at certain times of the year[8, 10, 13].

Measuring the flux of POC in situ at various depths in the water column is challenging, resulting in relatively poor spatial and temporal coverage of measurements when compared to more easily measureable ocean variables such as temperature or nitrate. Empirically derived models for POC flux that utilise limited in situ snapshot POC flux measurements may not capture episodic fluxes, such as those driven by krill swarms, and likely underestimate the carbon flux in regions of high krill density. This is especially true for models with a spatially and temporally invariant attenuation rate (typically Martin's *b* value[14]). This means that individual krill swarms will not be resolved either in time or space. Without mechanistic representation of the FP flux associated with krill swarms, biogeochemical models may not make accurate projections of the contribution of the Southern Ocean to global carbon export.

Antarctic krill have a life cycle strongly tied to seasonal sea ice, which provides nursery areas for the larvae[15] and can enhance local primary production as it melts, providing rich grazing grounds[16, 17]. The role of the marginal ice zone (MIZ; 15–80% ice cover[18]) as a feeding ground for high densities of krill[19], and the resulting high FP fluxes and low attenuation rates of sinking POC, suggest that the MIZ may be an area of significant POC export in the Southern Ocean[20, 21]. However, particle flux measurements are even more limited in the MIZ (in part due to the difficulties of sampling in sea ice) and, as such, the potential contribution of krill FPs in the MIZ to Southern Ocean export flux has yet to be fully quantified. Additionally, the patchy distribution of krill biomass[22] may mean that episodic events are poorly represented in many biogeochemical sampling campaigns. The combination of these factors mean that there are few observational data capturing these potentially large krill FP fluxes

(Supplementary Table 1). As empirical algorithms to estimate global export generally rely on extrapolation of datasets of in situ measurements[23–25], these episodic, but recurrent, krill FP fluxes are likely to be omitted, or at least under-represented, in Southern Ocean flux estimates.

We previously measured high krill FP fluxes and low attenuation rates in the MIZ of the South Orkney Islands[2], and a small number of other studies have also encountered large fluxes of krill FP associated with low attenuation rates through the upper mesopelagic[8, 11, 20, 26]. We hypothesise that, where krill densities are high in the MIZ, they will drive large FP fluxes such that the MIZ accounts for a substantial component of the total POC export in the Southern Ocean. To assess the contribution of krill FP to the BCP, we develop an empirical model to make first order estimates of this flux based on spatially-discrete krill density data over the past century (KRILLBASE[27]) and in situ measurements of krill FP attenuation in the MIZ of the Southern Ocean. We estimate that the seasonal flux of krill FPs at 100 m across the MIZ of the Southern Ocean can be large, and is equivalent to 17–61% of satellite-derived estimates of total carbon flux at this depth. These results highlight that krill FPs are an important contributor to carbon flux and yet they are not mechanistically represented in global biogeochemical models, which restricts our ability to predict future changes to the BCP.

## Results

**Krill faecal pellet production and export**. The total seasonal krill FPP over the MIZ region is estimated to be 0.065 GT C (ranging from 173 to 2427 mg C m$^{-2}$ over the season, with a mean of 786 mg C m$^{-2}$). FPP is highest in the Scotia-Weddell Sea area during the period 24 December–06 January (Fig. 1). This area, downstream of the Antarctic Peninsula, supports some of the highest krill densities in the Southern Ocean[15, 27, 28].

We conservatively estimate the export flux of krill FP carbon at 100 m (FP$_{100}$) based on the above predicted FPP and on a Martin type[14] attenuation curve with an attenuation coefficient of 0.32 (the median of literature estimates of krill FP attenuation valid for our region[2, 8, 9, 11, 12, 20]). Averaged over the entire MIZ area (FP$_{100,MIZ}$), highest FP export fluxes (104 mg C m$^{-2}$ d$^{-1}$) occur during the period 24 December–06 January. The total seasonal export of krill FP is highest in the Scotia and Weddell seas (peaking at 80 g C m$^{-2}$ in the MIZ region −35 to −40 °E). Summed over the productive season, we estimate the total export of krill FP at 100 m in the MIZ (FP$_{100,SEA}$) to be 0.039 GT C (Table 1).

**Model comparisons**. We compare our estimate of krill FP export to total POC export from a number of studies[23–25] that apply empirical algorithms for carbon export to satellite-derived primary productivity data (Table 1). If we assume that these algorithms correctly estimate the POC export in the MIZ, krill FP export could make up 17–61% (mean 35%, median 32%) of total seasonal POC export in the region south of the maximum MIZ extent (Table 1). Additionally, we compare our FP export estimates to three POC export models specific to the Southern Ocean (Table 1), which utilise nutrient and hydrographic data to constrain model parameters and budgets[29–31] (see methods). The estimate of POC export south of 50 °S by Schlitzer et al. (2002)[30] is much higher than other estimates, likely in part due to the inclusion of highly productive coastal waters off South America, and is therefore not used for further comparison. Based on the models of Primeau et al. (2013) and MacCready et al. (2001), which offer the best coverage of the Southern Ocean region analysed here, krill FP fluxes are 13–18% of the total annual POC export flux, compared to 14–43% for satellite-derived estimates for the region south of 60 °S (Table 1). These percentage

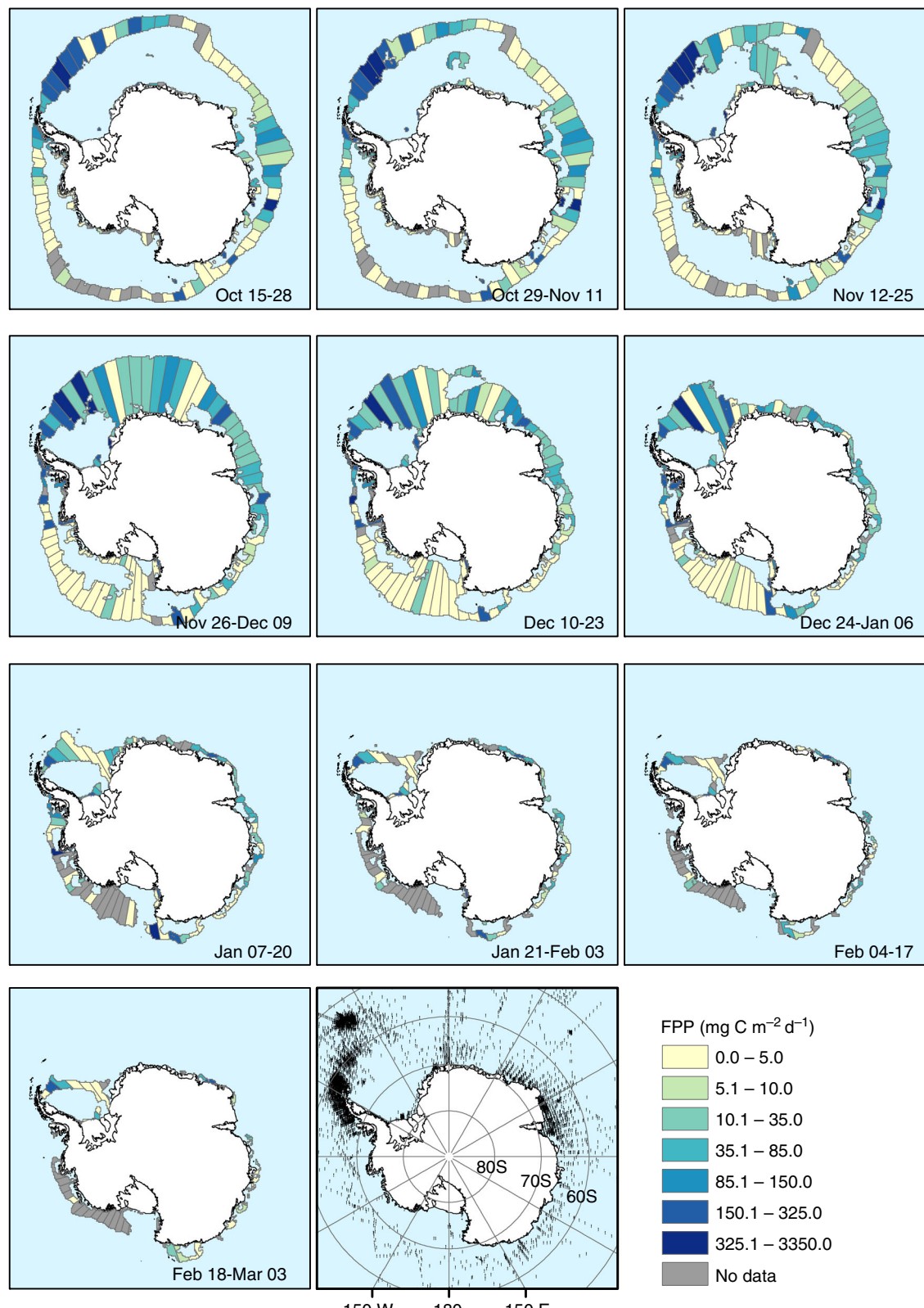

**Fig. 1** Estimated krill faecal pellet production (FPP) in the marginal ice zone, Antarctica. The marginal ice zone, from fortnightly sea ice concentration data (15–80% ice cover), is divided into 5 ° zonal cells, and is coloured by FPP (mg C m$^{-2}$ d$^{-1}$) based on krill density measurements from KRILLBASE (black dots) and literature krill FP production rates. Regions of the marginal ice zone where no KRILLBASE records occurred are coloured in grey. The Antarctic coastline was obtained from the SCAR Antarctic Digital Database

**Table 1 Comparison of krill faecal pellet (FP) POC export at 100 m (this study) with literature estimates of total POC export**

| Data source/model algorithm | POC export flux in the MIZ[a] (GT C yr$^{-1}$) | Krill FP contribution to POC export[b] (%) | POC export flux south of 60 °S[c] (GT C yr$^{-1}$) | Krill FP contribution to POC export[b] (%) |
|---|---|---|---|---|
| This study -krill FPs only | 0.039 | | | |
| *Satellite-based estimates*[d] | | | | |
| Carr, Henson | 0.076 | 51.5 | 0.101 | 38.5 |
| Carr, Dunne | 0.122 | 32.0 | 0.152 | 25.6 |
| Carr, Laws | 0.209 | 18.6 | 0.281 | 13.8 |
| Marra, Henson | 0.082 | 47.2 | 0.104 | 37.4 |
| Marra, Dunne | 0.137 | 28.4 | 0.160 | 24.3 |
| Marra, Laws | 0.227 | 17.1 | 0.289 | 13.5 |
| VGPM, Henson | 0.064 | 60.6 | 0.090 | 43.3 |
| VGPM, Dunne | 0.103 | 37.6 | 0.135 | 28.8 |
| VGPM, Laws | 0.177 | 21.9 | 0.250 | 15.5 |
| *POC export models* | | | | |
| Primeau et al. 2013[e,f] | | | 0.22 | 17.7 |
| Schlitzer et al. 2002[e] | | | 1.00 | 3.9 |
| MacCready et al. 2001[e] | | | 0.31 | 12.5 |

[a]Estimates are for the productive season, October–March. For satellite-based estimates these are for the region south of the maximum ice extent during the productive period defined here (i.e. location of 15% sea ice concentration for period 1–13 October 1994–2014)
[b]Percentage contribution of our estimate of krill FP export flux (0.039 GT C yr $^{-1}$) to various total POC export estimates
[c]Annual export for the region south of 60 °S, with the exception of Schlitzer et al. (2002) and Primeau et al. (2013) which are for the regions south of 50 °S and 55–60 °S, respectively.
[d]Named according to the input primary production and export (see methods)
[e]Annual export
[f]Export at base of euphotic zone (73.4 m)

contributions are conservative as our krill FP export flux is summed over the seasonal MIZ area, rather than south of 60 °S.

**Energy budget calculations.** We make independent estimates of krill FP flux at 100 m based on energy budget considerations. Taking estimates of circumpolar total gross krill production for krill of 20–50 mm in length (0.342–0.536 Gt yr$^{-1}$[32]), gross growth efficiencies of 20–30%[33], and absorption efficiencies (AE) of 0.42–0.94[4], we estimate FPP rates of 0.01–0.25 Gt C during our productive season (October–March), resulting in FP fluxes at 100 m of 0.007–0.150 Gt C yr$^{-1}$. This range is large, reflecting in particular the high range in AE, related to feeding rates and food type. Constraining this range in AE to 0.75–0.85 to reflect the more typical range of values reported[3], we estimate FP fluxes at 100 m of 0.016–0.065 Gt C yr$^{-1}$.

**Discussion**

Taking available data in the literature, we have made first order estimates of the flux of krill FP in the MIZ of the Southern Ocean. Our model highlights that the seasonal export of krill FP is highest in the Scotia and Weddell Seas, a reflection of both high krill densities and the large extent of the MIZ in this region in austral summer. According to the KRILLBASE statistical model[27, 28], peak krill densities occur during 24 December–06 January and consequently, highest FP export fluxes (104 mg C m$^{-2}$ d$^{-1}$ averaged over the entire MIZ area, FP$_{100,MIZ}$) are predicted to occur during this period. For comparison, maximum fluxes of cylindrical FPs of 125.5 mg C m$^{-2}$ d$^{-1}$ were observed in sediment traps at 170 m depth on the Western Antarctic Peninsula[10] in February. This agrees well with our calculated maximum krill FP flux of 121 mg C m$^{-2}$ d$^{-1}$ at 170 m in December in the region encompassing the Antarctic Peninsula (−80 to −60 °E), providing confidence in our model calculations.

We estimate that, over the productive season (October–March), the total export flux of krill FPs at 100 m in the MIZ (FP$_{100,SEA}$) is 0.039 GT C (Table 1). This represents 17–61% of satellite-based estimates of total carbon export, highlighting that krill FP export may represent a significant proportion of estimated total POC export flux for the MIZ. The incorporation

of POC export data collected via time-integrating methods, such as thorium (a radioactive tracer) as in the algorithms of Henson[23] and Dunne[25] (see methods), increases the chance that any carbon fluxes associated with ephemeral krill swarms are included in these satellite-based empirical algorithms. However, in situ POC export data are spatially limited and coverage in the MIZ is poor (Supplementary Figure 1); hence, these empirically derived estimates likely underrepresent the role of krill FP fluxes. If we assume that satellite-derived POC export estimates miss the contribution from krill FP in the MIZ modelled here, satellite-derived POC export estimates represent only 62–85% of the total POC export in these regions.

We estimate that krill FP fluxes represent 12.5–17.7% of total carbon export (Table 1) based on the models of MacCready et al. (2001) and Primeau et al. (2013). Krill FP fluxes could therefore represent a significant fraction of Southern Ocean POC export flux and their mechanistic inclusion in global models could improve projections of future ocean carbon uptake. The relatively sparse data availability of POC flux estimates in the Southern Ocean, and the lack of the mechanistic inclusion of krill FP export may lead to underestimations of the contribution of the Southern Ocean to global export fluxes.

There are a number of uncertainties associated with our estimates due to necessary assumptions and the degree to which input parameters are constrained by available data. Firstly, we take the FPP rate from literature for a standard krill of 600 mg fresh weight[3]. This is equivalent to a krill of length 34 mm based on the mass-to-length relationships calculated for the Scotia Sea in 2000[34]. Larger krill will produce larger FPs, which sink more rapidly and thus have a lower attenuation rate, where-as the opposite is true for smaller krill. The krill length of 34 mm sits at the low end of values reported in the field[28, 32, 35, 36], therefore our FPP rate is very conservative based on size.

Additionally, we assume a constant FPP rate throughout the day. It is likely that feeding rates, and thus egestion rates will change with food availability and season[4, 17], but this process is not yet sufficiently constrained to be incorporated into our model. If krill were only egesting for 12 h a day then we would over-estimate the flux of FPs by a factor of two. However, as the range in FP production estimates in the literature vary by over an order

**Table 2 Estimated FP export flux at 100 m for various sensitivity runs for the period October–March**

| Run | KRILLBASE data | MIZ data period | FPP rate (mg C ind$^{-1}$ d$^{-1}$) | Attenuation rate (Martin's $b$ value) | Total FP$_{100}$ export flux (GT C) | % of satellite export[a] |
|---|---|---|---|---|---|---|
| A | All | 1994–2014 | 3.2 | 0.32 | 0.039 | 17.1–60.6 |
| B | All | 1994–2014 | 0.67 | 0.32 | 0.008 | 3.7–13.1 |
| C | All | 1994–2014 | 6.29 | 0.32 | 0.079 | 34.6–122.8 |
| D | All | 1994–2014 | 3.2 | 0.10[b] | 0.055 | 24.3–86.3 |
| E | All | 1994–2014 | 3.2 | 0.62[b] | 0.024 | 10.5–37.3 |
| F | Unstandardised—median[c] | 1994–2014 | 3.2 | 0.32 | 0.002 | 0.9–3.1 |
| G | Unstandardised—90%[d] | 1994–2014 | 3.2 | 0.32 | 0.053 | 23.9–82.1 |
| H | Unstandardised—mean[e] | 1994–2014 | 3.2 | 0.32 | 0.031 | 13.9–49.0 |

A is the standard model run; the remaining runs are as for run A but with the following adjustments, B: minimum FPP rates, C: maximum FPP rates, D: minimum FP attenuation rates, E: maximum FP attenuation rates, F: Unstandardised KRILLBASE data –median density, G: Unstandardised KRILLBASE data –90th percentile density, H: Unstandardised KRILLBASE data –mean density
[a]Total krill FP export flux at 100 m (FP$_{100}$) as percentage of a range of satellite-derived estimates of total POC export (Table 1)
[b]Literature derived FP attenuation rates[8, 11]
[c]Calculated assuming a spatially and temporally constant krill density equal to the median of unstandardised krill density data >0 ind. m$^{-2}$
[d]Calculated assuming a spatially and temporally constant krill density equal to the ninetieth percentile value of unstandardised krill density data >0 ind. m$^{-2}$
[e]Calculated assuming a spatially and temporally constant krill density equal to the mean of all unstandardised krill density data

of magnitude,[2–4, 37, 38] any diel variation is small in comparison. We recalculate total krill FP$_{100}$ fluxes based on maximum (6.29 mg C ind.$^{-1}$ d$^{-1}$) and minimum (0.67 mg C ind.$^{-1}$ d$^{-1}$) FP production rates in the literature[3, 4, 37, 38]. This results in seasonally integrated FP production (FP$_{100,SEA}$) ranging between 0.008 and 0.079 GT C (sensitivity runs B and C, Table 2) highlighting the need for further studies on krill FPP rates to constrain this parameter or accurately model its variability (perhaps based on factors such as food availability). However, even with minimum rates of FPP, krill FP fluxes could still account for 4–13% of the total POC export flux based on satellite-derived estimates.

We also test the sensitivity of our model to the attenuation rate. The range in krill FP attenuation rates applicable to the Southern Ocean MIZ is 0.10–0.62[2, 8, 9, 11, 12, 20]. Using the upper and lower bounds of this range, we calculate FP$_{100,SEA}$ of 0.024 and 0.055 GT C respectively (sensitivity run D and E, Table 2). Even at the high end of literature derived attenuation rates, krill FP fluxes are 11–37% of the total POC export flux based on satellite-derived estimates, highlighting the potential magnitude of the missed POC flux. Additionally, the fact that our energy budget estimates of krill FP flux at 100 m (0.016–0.065 Gt C yr$^{-1}$) encompass our best model estimate (0.039 GT C yr$^{-1}$), gives additional confidence in our results. Therefore, although data are limited, the available literature supports the notion that krill FPs make an important contribution to the POC export flux in marginal ice zones, particularly those with high krill abundances.

Krill density values have been derived from KRILLBASE[27], in which net haul data have been standardised to a common sampling strategy to take into account varying levels of catch efficiency. However, as krill are able to escape nets, even with the most efficient net sampling strategy[39], our estimates of krill density are likely conservative. To assess the impact of the use of standardised KRILLBASE densities (for the subset of KRILLBASE data used in our study), we recalculate the seasonal FP flux at 100 m based on krill densities of 1.1 and 29.8 ind. m$^{-2}$ (sensitivity runs F and G, Table 2). These are the median and 90th percentile values for the unstandardised data, where densities are >0 ind. m$^{-2}$, which we believe to be a fair representation of the possible range of the mean Southern Ocean krill density, whilst not being heavily biased by zero values or rare extreme values. This results in FP$_{100,SEA}$ of 0.002 and 0.053 GT C (Table 2) based on median and 90th percentile values respectively, with the lower estimate likely to be at the extreme end. Taking the mean of all unstandardised krill density data (17.8 ind. m$^{-2}$) results in FP$_{100,SEA}$ of 0.031 GT C (sensitivity run H) which is close to our estimate of 0.039 GT C for standardised data. The use of standardised KRILLBASE data does therefore

not overly influence the conclusions drawn here. Since the marginal ice zone is harder to access and less well sampled than open waters, the KRILLBASE data are skewed towards regions of lower density, again increasing the likelihood that our estimates of krill FP flux are conservative.

Although our model results show that the estimated FP export flux in the MIZ is dominated by high krill density areas, this does not consider the effects of differences in zooplankton community structure, particularly in the abundance of microcopepods, many of which are coprorhexic or coprophagic. A number of studies have measured high retention of FPs in the euphotic zone, due most likely to copepod retention filters[40, 41], and currents generated by the swimming activities of both krill[42] and copepods[43] that could cause FP fragmentation, slow sinking rates and increased availability to smaller grazers such as dinoflagellates and ciliates[44]. Zooplankton community structure is therefore a key consideration when examining the degree to which sinking FPs are exported from the euphotic zone[21, 45–47]. Nevertheless, the importance of krill FPs in sediment traps (Supplementary Table 1)[2, 8, 9, 11, 12, 20], combined with evidence presented here, suggest that this retention filter could be short-circuited through a rain of large, fast sinking FPs[4, 9, 47], when krill densities (compared to microzooplankton grazers) are sufficiently high.

Our model of krill FP export flux represents a first-order estimation of the importance of krill for the export of carbon out of the euphotic zone of the MIZ of the Southern Ocean. We find krill FP export fluxes in the MIZ to be large which, if missed by current satellite-derived estimates of POC export (due to poor data availability in the MIZ), would add an additional 17–61% to POC export estimates for the Southern Ocean (Table 1), i.e. these empirical algorithms could underestimate POC export by 15–38%. There is a need for increased spatial and temporal coverage of export data in MIZ regions to allow better representation of these important regions in empirically derived global estimates of POC export. Additionally, our work highlights that krill FP fluxes need to be mechanistically represented in global biogeochemical models to enable more accurate projections of the future Southern Ocean contribution to carbon export. As a site of deep water formation and also a region where deep nutrient-rich waters are exposed to the atmosphere[48], the Southern Ocean is a key part of global biogeochemical cycles and contributes significantly to global export production[30]. It is therefore vital to quantify the efficiency of carbon transfer in the Southern Ocean as accurately as possible. Additionally our conclusions can be extended more globally, as the contribution to POC flux of FPs produced by other abundant swarming species, such as salps[49, 50],

may also be under-represented in particle flux studies depending on the decomposition state of FPs[51].

## Methods

**Krill faecal pellet production.** Fortnightly climatologies of the extent of the marginal ice zone (MIZ) in Antarctica were calculated from daily long-term passive microwave sea ice concentration data from 1994–2014 from the National Snow and Ice Data Center (NSIDC)[52]. Median sea ice concentrations were calculated over the productive season (October–March[53]), and the MIZ region defined as 15–80% ice cover[18]. The location and size of the MIZ can change rapidly[52] and therefore using fortnightly MIZ data reduces uncertainties associated with short timescale variability in MIZ areal extent. Each fortnightly MIZ region was further divided into 5° zonal cells (k). For each fortnightly MIZ region, average krill densities in each cell were estimated using data from KRILLBASE; a database compiling 15,194 net hauls taken in the Southern Ocean between 1926–2016[27]. Although KRILLBASE is the most comprehensive spatially-resolved dataset of Antarctic krill density, there are not sufficient data to cover each MIZ cell for every fortnight in the austral summer. Therefore, we take each spatially specific krill density data point from KRILLBASE and model the krill density at this location for each fortnightly period using an established model of krill density dynamics during the austral summer[27, 28]. In this way, we obtain better coverage over the MIZ for each fortnight (Supplementary Figure 2). This model, in which krill density increases from October to a maximum in early January, before decreasing again until March, has been used to standardise KRILLBASE density estimates to a relatively efficient sampling strategy (to a night-time RMT-8 net haul to 200 m on 1 January)[27, 28]. We use the other elements of this standardisation, i.e. to a night-time RMT-8 haul to 200 m, to control for differences in net size, sampling depth, and time of day[27, 28], each of which may affect the degree of undersampling due to net avoidance or vertical distribution patterns. Despite standardisation to an efficient net sampling strategy, net avoidance[39] means that even these standardised krill densities are likely an underestimate of true krill density. We used the haul and pooled stratified haul data, and excluded records representing survey means due to the potential for these survey means to cover more than one zonal cell. We also excluded data collected outside the austral spring and summer (October–March), or exclusively in deep strata. We converted each standardised krill density data point in KRILLBASE (with aforementioned exclusions) to a set of equivalent estimates for the mid-date of each fortnight using the standardisation model. We calculated $\bar{N}_{k,t}$, the average krill density in cell $k$ at time (fortnight) $t$, as the mean of all date-specific density estimates contained within the time-specific boundaries of cell $k$.

There are some caveats with our use of these data. Firstly, even though standardised to a single, relatively efficient sampling method, some have argued that issues of net mesh selection and avoidance will lead to underestimates of post-larval krill density[39]. This would lead to our estimates of krill faecal pellet production (FPP) being conservative. Secondly, we calculate krill densities in each MIZ zone by averaging all KRILLBASE data in that zone, creating a long-term average climatology of abundance. Therefore actual values in any given year will vary considerably about this mean[28]. Thirdly, our approach is sensitive to spatial differences in data availability (Supplementary Figure 2) with less reliable krill density estimates for sparsely sampled cells. Our use of the standardisation model to augment density estimates with data collected at other times of year was designed to reduce the impact of such spatial differences in data availability. However, this standardisation model does not represent any longitudinal differences in intra-annual krill dynamics, which could cause spatial differences in the accuracy of modelled dynamics. We also tested the sensitivity of our results to FPP rate and FP attenuation rate which demonstrate that our broad conclusions are robust (Table 2).

Fortnightly values of krill faecal pellet production (fpp) in each cell were calculated by multiplying mean krill densities (ind. m$^{-2}$) by a krill FPP rate (E) of 3.2 mg C ind.$^{-1}$ d$^{-1}$ from the literature[2, 3]. The average FPP for the circumpolar MIZ area (FPP$_{MIZ}$, mg C m$^{-2}$ d$^{-1}$) was calculated for each fortnightly period (t) using the following equation:

$$\text{fpp}_{MIZ,t} = \sum_{\text{MIZ}} \left( \bar{N}_{k,t} \times E \times A_{k,t} \right) / A_{MIZ,t}. \tag{1}$$

Here, $A_{k,t}$ is the area of the 5° cell bounded by the MIZ (m$^2$), $A_{MIZ,t}$ the total Southern Ocean MIZ area (m$^2$) for that fortnightly period excluding cells with no data, and $\bar{N}_{k,t}$ the mean krill density in that cell. Similarly, the total flux of FPs produced in the MIZ in each fortnightly period (fpp$_{TOT}$, mg C) was calculated using the following equation:

$$\text{fpp}_{TOT,t} = \sum_{\text{MIZ}} \left( \bar{N}_{k,t} \times E \times A_{k,t} \right) \times 14 \, \text{days}. \tag{2}$$

**Krill faecal pellet fluxes.** We model the export flux of FPs at depth $z$ (FP$_z$, mg C m$^{-2}$ d$^{-1}$) in each cell based on literature estimates of krill FP attenuation[2, 8, 9, 11, 12, 20] appropriate to our study region. We apply the median attenuation rate ($b = 0.32$) to the FPP data in each cell, using the Martin curve[14].

We calculate for $z = 100$ m (FP$_{100}$, mg C m$^{-2}$ d$^{-1}$) and, for comparison with a previous study[10], for $z = 170$ m.

$$\text{FP}_z = \text{fpp} \times (z/z_0)^{-b} \tag{3}$$

Here $z_0$ is the depth at which krill FPs are produced (taken as 20 m, based on mean swarm depths of 18.9 m measured in the southern Scotia Sea in spring[36]). We model the flux of FPs over the MIZ region (FP$_{100,MIZ}$, mg C m$^{-2}$ d$^{-1}$) and the total flux of FPs at 100 m in the MIZ for each fortnightly period (FP$_{100,TOT}$, mg C),

$$\text{FP}_{100,MIZ,t} = \sum_{\text{MIZ}} \left( \text{FP}_{100,t} \times A_{k,t} \right) / A_{MIZ,t}, \tag{4}$$

$$\text{FP}_{100,TOT,t} = \sum_{\text{MIZ}} \left( \text{FP}_{100,t} \times A_{k,t} \right) \times 14 \, \text{days}. \tag{5}$$

The total seasonal (October–March) FP$_{100}$ flux (FP$_{100,SEA}$, mg C) was calculated by summing the fluxes from each fortnightly period.

$$\text{FP}_{100,SEA} = \sum_{\text{Mar}}^{\text{Oct}} \left( \text{FP}_{100,TOT} \right) \tag{6}$$

**Model comparisons.** We compare our modelled krill FP export fluxes to a number of different modelled POC export fluxes at 100 m based on satellite-derived primary productivity estimates[54–56] and algorithms for export production[23–25] for the region south of the maximum ice extent during the productive period defined here (15% sea ice concentration during period 1–13 October 1994–2014) (Table 1). These export estimates use primary productivity (PP) estimates derived from satellite chlorophyll-a via three main algorithms, referred to here as Carr[56], Marra[55], and VGPM[54]. Satellite-based PP estimates have been shown to disagree with each other[57] and likely underestimate PP in the Southern Ocean[30, 58]. PP data are then used to calculate export production via three different algorithms, herein referred to as Laws[24], Dunne[25], and Henson[23]. The Laws algorithm utilises nitrate uptake data and f-ratio (the ratio of new production to total production); however, the validity of f-ratio derived export estimates was recently questioned[59] as they do not account for nitrification and therefore may overestimate export. In addition, only one of 11 samples used to derive the Laws algorithm is in the MIZ of the Antarctic. The Dunne algorithm uses estimates of export based on thorium, sediment traps, oxygen, and nitrate uptake data with 11 samples out of 122 in the MIZ. The Henson algorithm utilises solely thorium-derived particle export estimates, which represents time-integrated export estimated over the half life of thorium (24.1 days); 15 records out of 306 are in the MIZ. Because of its integration timescale, thorium-derived POC export estimates may better capture the passing of krill swarms and the associated high fluxes than sediment trap data. However, even if these high fluxes are captured by these time-integrated methods, or if in some MIZ regions large FP fluxes from krill swarms occur regularly enough to be captured by more short term particle flux methods, the limited number of data points in the MIZ used in the aforementioned algorithms (Supplementary Figure 1) means that the low attenuation rates occurring here will not be well represented. Only 5, 9, and 9% of data in the Henson, Laws, and Dunne algorithms, respectively, are from the MIZ of the Antarctic. This highlights the need for more observations in the MIZ to support regionally derived estimates of particle flux attenuation, and the incorporation of these into global export flux estimates.

In addition we compare our values of krill FP export to three biogeochemical models estimating POC export in the Southern Ocean (Table 1)[29–31]. Estimates of Primeau et al. (2013) are based on a data-assimilating (temperature, salinity, and radiocarbon distributions) model (horizontal resolution of 2 x 2°) of ocean phosphate, calculated for their Antarctic zone (~south of 55–60 °S) as defined by latitude of maximum Ekman divergence[31]. MacCready et al. (2001) use a diagnostic model (with zonally averaged 5° bins) to calculate the physical fluxes in the surface nitrate budget, using a climatology of nitrate concentration (from historical CTD and bottle data primarily from WOCE and US JGOFS) to constrain nitrogen export[29]. Schlitzer et al. (2002) estimate carbon export south of 50 °S from an inverse model based on hydrographic, nutrient (accounting for nitrification), oxygen, and carbon data, with horizontal resolution ranging from 5 × 4° longitude by latitude in open ocean areas to 2.5° to 1° in regions with narrow currents[30]. These regional models are not reliant on snapshot style POC flux measurements and provide better spatial coverage of the MIZ than satellite-based estimates.

**Energy budget calculations.** To obtain an independent estimate of krill FP fluxes, we take estimates of krill production and apply energy budget considerations. We take the estimate of circumpolar krill production of 0.342–0.536 Gt yr$^{-1}$ for krill 20–50 mm in length[32], which is based on KRILLBASE krill densities and an empirical model[35] based on krill length, food concentration (from SeaWiFS satellite chlorophyll) and sea surface temperature (MODIS satellite). This is the production during a 4-month summer from 01 December to 31 March[32]. Converting this wet mass production to units of carbon, we assume dry mass is 25% of wet mass[32] and

carbon content is 43% of dry mass[4]. Total carbon ingestion ($C_{ing}$) was then estimated based on gross growth efficiencies of 20–30%[33], and FP production (fpp) calculated using absorption efficiencies ($a$) of 0.42–0.94[4] and the following relationship.

$$C_{ing} = fpp/(1 - a) \qquad (7)$$

We also estimate fpp using a more constrained range in $a$ of 0.75–0.85[3]. As before, we use equation 3 (above) to estimate the krill FP flux at 100 m, and sum over our productive season of (October–March).

## Data availability

The KRILLBASE dataset analysed for this study is available online: A. Atkinson, S. L. Hill, E.A. Pakhomov, V. Siegel et al. (2016). KRILLBASE: A database of Antarctic krill and salp densities in the Southern Ocean, 1926 to 2016. [http://doi.org/brg8]. The marginal ice zone data were extracted from the National Snow and Ice Data Center, available here: Cavalieri, D. J., Parkinson, C. L., Gloersen, P. & H. J. Zwally. Sea Ice Concentrations from Nimbus-7 SMMR and DMSP SSM/I-SSMIS Passive Microwave Data. *Boulder, Colorado USA. NASA National Snow and Ice Data Center Distributed Active Archive Center* (1996) [https://doi.org/10.5067/8GQ8LZQVL0VL]. The final data generated in this study are contained within Tables 1 and 2 in this manuscript, with supplementary data and figures as a Supplementary Information file.

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

## Acknowledgements

We would like to thank all the crew and scientists involved in collecting the decades of krill density data used to create the KRILLBASE database. Anna Belcher, Geraint A. Tarling, Clara Manno, Sally E. Thorpe, and Simeon L. Hill were supported by the Ocean Ecosystems programme at British Antarctic Survey and a Large Grant from the UK Natural Environment Research Council (NE/M020835/1). Stephanie Henson was supported by a European Research Council Consolidator grant (GOCART, agreement number 724416). Angus Atkinson was funded through the UK Natural Environment Research Council through its National Capability grant number (NE/R015953/1). Additionally Anna Belcher was supported by a PhD studentship from the University of Southampton.

## Author contributions

A.B., G.A.T., and C.M. derived early hypotheses, A.B and G.A.T. designed the analytical approach and A.B. analysed results. P.F., L.I., and S.E.T. provided MIZ data analysis support. A.A. helped with KRILLBASE and energy budget calculations. S.A.H. conducted model comparisons and S.L.H. estimated fortnightly krill densities. A.B. wrote the manuscript with contributions by all co-authors.
