## [Peer Review File · Nature Communications]

Reviewers' comments:

Reviewer #1 (Remarks to the Author):

The manuscript assesses the importance of intense pulses of sinking krill fecal pellets as a contributor to the carbon pump in the marginal ice zone of the Southern Ocean. The authors estimate that 0.039 GT C/yr is exported by krill fecal pellets in episodic pulses, which corresponds to 35% (27-61%) of the satellite-derived export estimate.

The manuscript is clearly written and the results are convincing. The only suggestion I have is that the authors add a bit more detail describing the types of models they are referring to when they state

on line 24 pg 1, "which is likely not captured by many global biogeochemical models"

and

on line 39-42 pg 2 "Global biogeochemical models that use temporally and spatially invariant attenuation rates (most notably Martin's b value) are therefore likely to underestimate the carbon flux in regions of high krill density",

because some (maybe all) global Earth System Models (ESMs) calibrate the remineralization length scales so that the models' control runs match observed vertical profiles of phosphate, nitrate, and oxygen. Thus, to the extent that krill fecal pellets are an important contributor to the biological pump, their climatological impact should be crudely captured by the ESMs. This of course does not mean that the ESMs capture the impact of Krill fecal pellets accurately or in a mechanistic way. Nevertheless, it is not clear that in the case of ESMs, adding a parameterization of Krill fecal pellet export should lead to an increase in carbon export. For the diagnostic satellite-derived export models on the other hand, I think the authors are correct that the missing impact of episodic and intense pulses of krill fecal pellets probably implies an under-estimation of the carbon export.

Other than that. I think the manuscript is a valuable contribution to the literature and should be published in Nature Communications.

Reviewer #2 (Remarks to the Author):

Review of the manuscript "Krill faecal pellets in the marginal ice zone: Hidden pulses of particulate organic carbon" by Belcher et al. (NCOMMS-18-23863-T).

The manuscript by Belcher et al. reports on the contribution of episodic faecal pellet carbon export events in the Marginal Ice Zone (MIZ) mediated by large krill swarms for the Southern Ocean Biological Carbon Pump (SO-BCP). The authors model this "hidden" flux based on literature data of faecal pellet production (FPP) and flux attenuation estimates as well as independent estimates of energy budget considerations and krill density data from KRILLBASE and compare their derived krill faecal pellet export with estimates from satellite and biogeochemical models.

Although the authors acknowledge the inherent limitations of their extrapolations given the scarcity and large spread of FPP and flux attenuation estimates as well as energy budget data for krill, the authors could have done a better job in outlining the uncertainties in their estimates (see detailed comments). Not being familiar with KRILLBASE myself, I furthermore wonder how different criteria and/or statistical approaches applied to the krill density data might impact the results? At least I was intrigued to read that a recent study by Cox et al. 2018 (No evidence for a decline in the density of Antarctic krill *Euphausia superba* Dana, 1850, in the Southwest Atlantic sector between 1976 and 2016. *Journal of Crustacean Biology* (2018) 1–6. doi:10.1093/jcbiol/ryy072) came to a very different conclusion than Atkinson et al. 2004 regarding the decline in krill stocks in the southwest Atlantic sector applying a different analysis of data extracted from KRILLBASE.

Overall the authors have to provide more compelling evidence that their modelling exercise is indeed robust enough to make their claims.

Detailed comments:

- Lines 43-45: The study by Meyer et al. 2018 (The winter pack-ice zone provides a sheltered but food-poor habitat for larval Antarctic krill. *Nature Ecology & Evolution*, <https://doi.org/10.1038/s41559-017-0368-3>) should be cited in this context.
- Lines 71-72: Are all post-larval krill assumed equal? Were any distinctions made between different post-larval krill size classes in terms of FPP when using the krill densities for upscaling? Also faecal pellets of smaller krill will have lower sinking rates.

- Lines 82-83: Comparing Figure 1 and Figure S2, data are quite often not very evenly distributed in some areas during some times of the year. Is it valid to extrapolate the krill FP export to the entire MIZ of the respective 5° cells and does this really constitute a conservative estimate, given that krill swarms are very patchy and likely more prominent at the ice edge or the part of the MIZ with less dense sea ice? How does krill FP export in the pack-ice zone and in open waters compare to the MIZ?
- Lines 100-103: Which studies include Thorium-based estimates?
- Line 123: I don't quite see the issue of including productive waters off South Georgia since it is an important krill area.
- Table 1: The last entry (12.5 from MacCreedy et al. 2001) seems to have slipped one line
- Line 150: I assume it should be seasonally integrated FP flux instead of FP production?
- Lines 186-188: Protozooplankton, in particular dinoflagellates and ciliates, could be key degraders of krill faecal pellets as has been shown for copepod faecal pellets (Poulsen, L. K. and M. H. Iversen (2008). "Degradation of copepod fecal pellets: key role of protozooplankton." Marine Ecology-Progress Series 367: 1-13.)
- Line 207: Do salps make the kind of large aggregations like krill resulting in faecal pellet "rain events" that would overwhelm the detrital feeders?
- Lines 227-229: To what extent does standardization to near maximum densities by upward corrected daytime hauls affect the FP export estimates? And is it valid to assume that krill will be feeding at the same rate during day as during night? Although from winter and for larval krill, the study of Meyer et al. 2018 indicates that krill seek shelter in sea ice during the day and have their main feeding activity at night when they swarm into the water column.
- Lines 237-240: To what extent is the undersampling of postlarval krill compensated by upwards corrected daytime hauls, extrapolation of uneven data to large areas and applying the same flux attenuation/FPP to all post-larval krill despite size differences?

Reviewers' comments:

Reviewer #1 (Remarks to the Author):

The manuscript assesses the importance of intense pulses of sinking krill fecal pellets as a contributor to the carbon pump in the marginal ice zone of the Southern Ocean. The authors estimate that 0.039 GT C/yr is exported by krill fecal pellets in episodic pulses, which corresponds to 35% (27-61%) of the satellite-derived export estimate.

The manuscript is clearly written and the results are convincing. The only suggestion I have is that the authors add a bit more detail describing the types of models they are referring to when they state

on line 24 pg 1, "which is likely not captured by many global biogeochemical models"

and

on line 39-42 pg 2 "Global biogeochemical models that use temporally and spatially invariant attenuation rates (most notably Martin's b value) are therefore likely to underestimate the carbon flux in regions of high krill density",

because some (maybe all) global Earth System Models (ESMs) calibrate the remineralization length scales so that the models' control runs match observed vertical profiles of phosphate, nitrate, and oxygen. Thus, to the extent that krill fecal pellets are an important contributor to the biological pump, their climatological impact should be crudely captured by the ESMs. This of course does not mean that the ESMs capture the impact of Krill fecal pellets accurately or in a mechanistic way. Nevertheless, it is not clear that in the case of ESMs, adding a parameterization of Krill fecal pellet export should lead to an increase in carbon export. For the diagnostic satellite-derived export models on the other hand, I think the authors are correct that the missing impact of episodic and intense pulses of krill fecal pellets probably implies an under-estimation of the carbon export.

Other than that. I think the manuscript is a valuable contribution to the literature and should be published in Nature Communications.

REPLY: Thank you for taking the time to read our manuscript and for the valuable comments. You

are right in that models tuned to profiles of phosphate, nitrate and oxygen may be capturing the effect of krill faecal pellet fluxes. However, the phosphate, nutrient and oxygen profiles that a model may be calibrated or validated against are typically the World Ocean Atlas climatological monthly values which are at 1° spatial resolution. These data are unlikely to fully capture episodic krill pulses, or at least will substantially underestimate their impact on nutrient distributions. Of the time series stations in the World Ocean Atlas data set, only one is in the Southern Ocean, and thus is not well represented. Therefore even ESMs that are tuned to World Ocean Atlas nutrient profiles are likely to be missing the full influence of episodic events. There is not scope in the abstract to convey these nuances of biogeochemical models, and thus we keep the mention of models succinct in the abstract:

‘The magnitude of our conservatively estimated flux, which is likely not captured by many global biogeochemical models,’

We have amended the text in the introduction to clarify to what degree different models may capture these fluxes, lines 50-68.

‘Measuring the flux of POC *in situ* at various depths in the water column is challenging, resulting in relatively poor spatial and temporal coverage of measurements when compared to more easily measurable ocean variables such as temperature or nitrate. Empirically derived models for POC flux that utilise limited *in situ* ‘snapshot’ carbon flux measurements may not capture episodic fluxes, such as those driven by krill swarms, and likely underestimate the carbon flux in regions of high krill density. This is especially true for models with a spatially and temporally invariant attenuation rate (typically Martin’s *b* value (Martin et al., 1987)), but episodic fluxes are likely also underrepresented in more sophisticated global biogeochemical models that use temporally and spatially varying attenuation rates. The remineralisation of krill faecal pellets at depth will impact nutrient profiles for which there is better global data coverage than for flux estimates. As some global biogeochemical models tune or validate the remineralisation length scale using profiles of nutrients such as phosphate and nitrate, it could be argued that these models implicitly incorporate episodic fluxes since the observed nutrient profiles capture the contribution from the remineralisation of krill FP at depth. However, the nutrient data used for validation are typically monthly mean climatologies at low spatial resolution (e.g. the 1° World Ocean Atlas) that, in the Southern Ocean, are frequently derived from sparse observations. This means that individual krill swarms will not be resolved either in time or space and therefore, these data, and hence the models, are also unlikely to capture episodic krill FP fluxes.’

Additionally in the discussion section we have revised our wording to make our intentions clear (lines 187-196)

‘The biogeochemical models may be better at capturing episodic pulses of krill FPs as they are tuned to hydrographic and nutrient observations, for which data availability is greater. However, the World Ocean Atlas data typically used for model calibration/validation is a monthly climatology at 1° spatial resolution, and incorporates only one time series station in the Southern Ocean. The temporal resolution and data coverage in regions of high krill density are therefore not adequate to capture episodic fluxes of krill FP fully. If we assume that the models of Primeau et al. (2013) and MacCreedy et al. (2001) omit the entire flux of krill FPs, these models capture 85-89 % of the total POC export south of 60 °S. Krill FP fluxes could therefore represent a significant fraction of Southern Ocean POC

export flux which may not be captured by global models.'

Reviewer #2 (Remarks to the Author):

Review of the manuscript "Krill faecal pellets in the marginal ice zone: Hidden pulses of particulate organic carbon" by Belcher et al. (NCOMMS-18-23863-T).

The manuscript by Belcher et al. reports on the contribution of episodic faecal pellet carbon export events in the Marginal Ice Zone (MIZ) mediated by large krill swarms for the Southern Ocean Biological Carbon Pump (SO-BCP). The authors model this "hidden" flux based on literature data of faecal pellet production (FPP) and flux attenuation estimates as well as independent estimates of energy budget considerations and krill density data from KRILLBASE and compare their derived krill faecal pellet export with estimates from satellite and biogeochemical models.

Although the authors acknowledge the inherent limitations of their extrapolations given the scarcity and large spread of FPP and flux attenuation estimates as well as energy budget data for krill, the authors could have done a better job in outlining the uncertainties in their estimates (see detailed comments). Not being familiar with KRILLBASE myself, I furthermore wonder how different criteria and/or statistical approaches applied to the krill density data might impact the results? At least I was intrigued to read that a recent study by Cox et al. 2018 (No evidence for a decline in the density of Antarctic krill *Euphausia superba* Dana, 1850, in the Southwest Atlantic sector between 1976 and 2016. *Journal of Crustacean Biology* (2018) 1–6. doi:10.1093/jcbiol/ruy072) came to a very different conclusion than Atkinson et al. 2004 regarding the decline in krill stocks in the southwest Atlantic sector applying a different analysis of data extracted from KRILLBASE.

Overall the authors have to provide more compelling evidence that their modelling exercise is indeed robust enough to make their claims.

REPLY: Thank you for taking the time to review our manuscript and for highlighting the need for clearer description of the uncertainties. We have responded to your comments below, and additionally have altered text in the methods section (shown below, line 310-315) to clarify the standardisation method of the KRILLBASE dataset. We stress that due to problems of net avoidance, nets typically underestimate the biomass of fast swimmers such as euphausiids and micronekton. Even with standardisation to the most efficient net sampling regime, estimates are still likely an underestimate of the true biomass.

'We use the other elements of this standardisation, i.e. to a night-time RMT-8 haul to 200 m, to control for differences in net size, sampling depth, and time of day (Atkinson et al., 2008, 2017), each of which may affect the degree of undersampling due to net avoidance or vertical distribution patterns. Despite standardisation to an efficient net sampling strategy, net avoidance (Everson and Bone, 1986) means that even these standardised krill densities are likely an underestimate of true krill density.'

To address concerns about the effect of using standardised KRILLBASE data, for each fortnightly period we have calculated the mean krill density of the standardised data (18-41 ind. m⁻²) based on the subset of KRILLBASE data used in our study, i.e. with stratified hauls etc removed, and before we

recalculate densities in only the MIZ. We then compare these to the mean krill density of the unstandardised data (18 ind. m⁻²). We find that the standardisation method employed increases krill density by a factor of 1.8 ± 0.4 from unstandardised values. This is still likely an underestimate of krill density for the aforementioned regions, and is also small when compared to the order of magnitude range in literature FP production rates.

With regard to the paper of Cox et al., (2018), they replace KRILLBASE standardisation with the broadly similar approach of including net type, time of day and time of year as explanatory variables in their models. Unlike KRILLBASE standardisation, they do not consider the effect of sampling depth, which is an important influence on krill density (e.g. Atkinson et al., 2017). The analysis of Cox et al (2018) uses only data from midwater trawls. Atkinson et al., (2004, their table 1) analysed a similar restricted dataset and did not find a significant trend. Thus where the analyses in Cox et al (2018) and Atkinson et al (2004) are comparable, they yield similar results. Like Atkinson et al (2004) we used other net types to extend our data coverage. We also controlled for the effects of net type using standardisation.

Estimates of carbon flux are sensitive to input values, including krill density. While our approach provides a defensible best estimate of krill density, we have explored this thoroughly through sensitivity analyses, which we have extended based on the suggestions of this review (Table 2), and have explained in detail in our responses below. We conclude that the FP production rate is responsible for the largest range in uncertainty and stress this as an area for future research to further constrain krill FP fluxes.

Detailed comments:

- Lines 43-45: The study by Meyer et al. 2018 (The winter pack-ice zone provides a sheltered but food-poor habitat for larval Antarctic krill. *Nature Ecology & Evolution*, <https://doi.org/10.1038/s41559-017-0368-3>) should be cited in this context.

REPLY: The citation has been added on lines 69-71.

- Lines 71-72: Are all post-larval krill assumed equal? Were any distinctions made between different post-larval krill size classes in terms of FPP when using the krill densities for upscaling? Also faecal pellets of smaller krill will have lower sinking rates.

REPLY: In this first order estimate of krill FP export flux we used a fixed faecal pellet production (FPP) rate which is based on a 'standard krill' of 600 mg fresh weight (Clarke et al., 1988). Fortnightly resolved krill length-frequency distributions for the entire Southern Ocean are not available, and as FPP rate also varies with factors such as food concentration and feeding rates (Atkinson et al., 2012), it would introduce additional uncertainties to model FPP rate based on krill size alone. Therefore to match the FPP rate of Clarke et al., (1988) we assume all krill to be 600 mg wet weight. This is equivalent to a 34 mm length krill, (Hewitt et al., 2004) and lies at the low end of measurements made in the field (Atkinson et al., 2006, 2009; Fielding et al., 2012). Therefore our estimates of FPP are likely conservative based on the size of krill used. We have added the following lines to the manuscript (lines 197-204) to make sure that this assumption is made clear.

'There are a number of uncertainties associated with our estimates due to necessary assumptions and the degree to which input parameters are constrained by available data. Firstly, we take the FPP rate from literature for a 'standard krill' of 600 mg fresh weight (Clarke et al., 1988). This is equivalent to a krill of length 34 mm based on the mass-to-length relationships calculated for the Scotia Sea in 2000 (Hewitt et al., 2004). Larger krill will produce larger FPs, which sink more rapidly and thus have a lower attenuation rate, where-as the opposite is true for smaller krill. The krill length of 34 mm sits at the low end of values reported in the field (Atkinson et al., 2006, 2009; Fielding et al., 2012), therefore our FPP rate is conservative based on size.'

The attenuation rates that we use are not tied to our estimate of FPP, and are representative of the range of krill FP sizes captured in the literature (and resultantly a range in krill sizes). Data on krill FP fluxes are typically available as total carbon rather than individual FP lengths, so we do not have the FP size data to accompany our calculated attenuation rates. However, as we have calculated the median krill FP attenuation rate based on a range of studies and across a number of different regions in the Southern Ocean, we believe that our attenuation rate is representative of size distribution of krill FPs occurring in the field. Additionally, lower attenuation rates of larger than average FPs are likely compensated for by higher attenuation rates of smaller FPs.

- Lines 82-83: Comparing Figure 1 and Figure S2, data are quite often not very evenly distributed in some areas during some times of the year. Is it valid to extrapolate the krill FP export to the entire MIZ of the respective 5° cells and does this really constitute a conservative estimate, given that krill swarms are very patchy and likely more prominent at the ice edge or the part of the MIZ with less dense sea ice? How does krill FP export in the pack-ice zone and in open waters compare to the MIZ?

REPLY: Although KRILLBASE is the most comprehensive compilation of krill net haul data to date, the data are indeed not always evenly spread in a particular 5° cell. In the methods (lines 330-333) we acknowledge the limitation of uneven data.

'Thirdly, our approach is sensitive to spatial differences in data availability (Supplementary Figure 2) with less reliable krill density estimates for sparsely sampled cells. Our use of the standardisation model to augment density estimates with data collected at other times of year was designed to reduce the impact of such spatial differences in data availability.'

The size of the cell was chosen for consistency with KRILLBASE (Atkinson et al., 2008), and for a balance in computational processing required. As the KRILLBASE density data are heavily skewed towards zero values, and as 90% of the standardised KRILLBASE data have density $<34 \text{ ind. m}^{-2}$, any extrapolation over a cell area will likely result in an underestimation of krill FP fluxes, and estimates of krill density in a cell are thus conservative.

The studies of Cadée et al., (1992) and Cadée, (1992) in the Scotia and Weddell Seas find high particulate organic carbon fluxes associated with the ice edge, with krill FPs dominating sediment trap material in the melting ice zone. Additionally González, (1992) noted that heavy krill grazing occurs during and following the ice retreat in the Weddell-Scotia Confluence. These observations likely result from high densities of krill and krill swarms in these regions. Since the marginal ice zone is harder to access and more undersampled than open waters, the KRILLBASE data itself is skewed

towards regions of lower density. We add the following text to the manuscript to highlight this (lines 258-261):

'Since the marginal ice zone is harder to access and less well sampled than open waters, the KRILLBASE data are skewed towards regions of lower density, again increasing the likelihood that our estimates of krill FP flux are conservative.'

Our model assumes that high FP export is directly related to high krill densities. We have added a sentence at the start of the results section to make this clear (lines 98-100).

'Our estimates of FPP and subsequently FP fluxes therefore directly relate to krill densities, with higher FPP production predicted in regions of high krill density.'

- Lines 100-103: Which studies include Thorium-based estimates?

REPLY: We have amended the text as follows (lines 175-179):

'The incorporation of POC export data collected via time-integrating methods, such as thorium (a radioactive tracer) as in the algorithms of Henson (Henson et al., 2011) and Dunne (Dunne et al., 2005) (see methods), increases the chance that any carbon fluxes associated with ephemeral krill swarms are included in these empirical algorithms.'

A full description of the data used by each of the algorithms that we use in this study is found in the methods section (lines 364-390).

- Line 123: I don't quite see the issue of including productive waters off South Georgia since it is an important krill area.

REPLY: Indeed, you are right, South Georgia is an important area for krill. It is the inclusion of the highly productive waters off South America that results in this model not being suitable for comparison. We have amended the text to clarify this, and mention only the productive waters off South America as the reason for not considering the model further (lines 129-131).

- Table 1: The last entry (12.5 from MacCreedy et al. 2001) seems to have slipped one line

REPLY: Thank you for spotting this, we have corrected it.

- Line 150: I assume it should be seasonally integrated FP flux instead of FP production?

REPLY: Amended

- Lines 186-188: Protozooplankton, in particular dinoflagellates and ciliates, could be key degraders of krill faecal pellets as has been shown for copepod faecal pellets (Poulsen, L. K. and M. H. Iversen (2008). "Degradation of copepod fecal pellets: key role of protozooplankton." Marine Ecology-Progress Series 367: 1-13.)

REPLY: We have amended lines 265-268 to include the impact of protozooplankton:

'A number of studies have measured high retention of FPs in the euphotic zone, due most likely to copepod retention filters (Wexels-Riser et al., 2001; Wexels Riser et al., 2007), and currents generated by the swimming activities of both krill (Dilling and Alldredge, 2000) and copepods (Poulsen and Kiørboe, 2005) that could cause FP fragmentation, slow sinking rates and increased availability to smaller grazers such as dinoflagellates and ciliates (Poulsen and Iversen, 2008).'

- Line 207: Do salps make the kind of large aggregations like krill resulting in faecal pellet "rain events" that would overwhelm the detrital feeders?

REPLY: Salps can indeed form large swarms (Martin et al., 2017; Ramaswamy et al., 2005; Smith, Jr. et al., 2014; Stone and Steinberg, 2014 and refs within) and can result in episodic pulses of carbon to the seafloor (Smith, Jr. et al., 2014). Given that salps graze at high rates and produce rapidly sinking FPs (Stone and Steinberg, 2016), it is likely that during such bloom events, large numbers of FPs will escape detrital feeders and be exported out of the upper ocean. However, it is important to consider that salp FPs can be fragile and that sinking velocities vary with decomposition state (Iversen et al., 2017). As with krill, salps may therefore be important for biogeochemical cycling. We have added further appropriate references in the text to support our statement. (lines 287-290).

- Lines 227-229: To what extent does standardization to near maximum densities by upward corrected daytime hauls affect the FP export estimates? And is it valid to assume that krill will be feeding at the same rate during day as during night? Although from winter and for larval krill, the study of Meyer et al. 2018 indicates that krill seek shelter in sea ice during the day and have their main feeding activity at night when they swarm into the water column.

REPLY: The standardisation methods of KRILLBASE are to account for the differing catchabilities of krill using various net types, as well as the effect of reduced catch efficiency during the daytime. As net avoidance is particularly pronounced during the day (e.g. Everson and Bone, 1986), daytime haul data will underestimate the abundance of krill in the water column. Catch efficiencies are a concern for animals with fast swimming abilities and it is not uncommon for correction factors to be applied. For example Ariza et al (2015) applied catch efficiencies of 80 % to euphausiid catch data based on the acoustic work of Davidson et al. (2011). We consider the standardised data to be a more accurate estimate of the krill biomass. Nevertheless, we have added additional sensitivity analysis to investigate the use of standardised krill data (lines 244-258 and updated Table 2).

'Krill density values have been derived from KRILLBASE (Atkinson et al., 2017), in which net haul data have been standardised to a common sampling strategy to take into account varying levels of catch efficiency. However, as krill are able to escape nets, even with the most efficient net sampling strategy (Everson and Bone, 1986), our estimates of krill density are likely conservative. To assess the impact of the use of standardised KRILLBASE densities (for the subset of KRILLBASE data used in our study), we recalculate the seasonal FP flux at 100 m based on krill densities of 1.1 and 29.8 ind. m⁻² (sensitivity runs F and G, Table 2. These are the median and 90th percentile values for the unstandardised data, where densities are > 0 ind. m⁻², which we believe to be a fair representation of the possible range of the mean Southern Ocean krill density, whilst not being heavily biased by zero values or rare extreme values. This results in FP_{100,SEA} of 0.002 and 0.053 GT C (Table 2) based on median and 90th percentile values respectively, with the lower estimate likely to be at the extreme end. Taking the mean of all unstandardised krill density data (17.8 ind. m⁻²) results in FP_{100,SEA} of

0.031 GT C (sensitivity run H) which is close to our estimate of 0.039 GT C for standardised data. The use of standardised KRILLBASE data does therefore not overly influence the conclusions drawn here.'

The FPP rate we use follows the assumption in Clarke et al., (1988) that 'an observed hourly rate can be multiplied by 24 to obtain an estimated daily rate'. Any diel changes in feeding rates and thus FPP rates would impact the flux of FP to the water column. However, this is not yet well constrained enough to be accurately incorporated into our model. The duration and rate of krill feeding (and thus egestion) will be affected not only by light levels, which change seasonally throughout the year, but also by food availability etc. We agree that feeding rates, and thus egestion rates, will vary throughout the day, but believe that the potential reduction by a factor of 2 (i.e. if krill were only feeding for 12 hours a day) is more than accounted for in our sensitivity analysis where we vary FPP by an order of magnitude from 0.67 to 6.29 mg C m⁻² d⁻¹. We agree that it is important to make clear our assumption of 24 hours feeding, and to discuss possible diel variability, thus we have added the following text (lines 205-210).

'Additionally we assume a constant FPP rate throughout the day. It is likely that feeding rates, and thus egestion rates will change with food availability and season (Atkinson et al., 2012; Meyer et al., 2017), but this process is not yet sufficiently constrained to be incorporated into our model. If krill were only egesting for 12 hours a day then we would over estimate the flux of FP by a factor of two. However, as the range in FP production estimates in the literature is over an order of magnitude, any diel variation is small in comparison.'

Additionally, to avoid overestimations of krill density, we do not use the KRILLBASE standardisation to maximum densities on January 1st. Instead we have used the KRILLBASE model of krill density dynamics to scale krill densities to each fortnightly period. In this way we model an increase in density from October to January, followed by a decline to March. We describe this in methods lines 304-310.

'we take each spatially specific krill density data point from KRILLBASE and model the krill density at this location for each fortnightly period using an established model of krill density dynamics during the austral summer (Atkinson et al., 2008, 2017). In this way we obtain better coverage over the MIZ for each fortnight (Supplementary Figure 1). This model, in which density increases from October to a maximum in early January, before decreasing again until March was used to standardise KRILLBASE density estimates to a relatively efficient sampling strategy (to a night time RMT-8 net haul to 200 m on January 1st) (Atkinson et al., 2008, 2017).'

• Lines 237-240: To what extent is the undersampling of postlarval krill compensated by upwards corrected daytime hauls, extrapolation of uneven data to large areas and applying the same flux attenuation/FPP to all post-larval krill despite size differences?

REPLY: These are important points and highlight the current uncertainties in krill biomass, egestion and attenuation rate. These are major issues to be tackled within our field. We do not present our study as the definitive estimate of krill FP fluxes in the Southern Ocean. Our aim is to provide a first order estimate of this, and to highlight the key assumptions and uncertainties that need to be addressed in order to constrain these estimates further. As discussed in our responses above, we

have added additional text to the manuscript and have conducted further sensitivity analyses to make the assumptions clear. We have revised the manuscript to make sure that these points come across and that we are not overstating our results.

It is likely that some of the uncertainties mentioned compensate for each other, however we are not able to quantify this based on current data available in the literature. As discussed above we believe that the extrapolation of uneven data, and poor net capture efficiencies result in an underestimation of the krill FP flux. The upward correction of daytime hauls in KRILLBASE is likely conservative. In our response above we explain why we believe that size differences in krill do not impact our attenuation rates, which are representative of a broad range of sites and seasons so are likely a good mean estimate. Thus, the overall effect of the above assumptions is an underestimate of the krill FP flux, and we have clarified this point in the text. The biggest uncertainty is the rate of egestion, for which literature estimates vary over an order of magnitude, likely related to feeding conditions etc. In lines 210- 215 in the manuscript we highlight this large uncertainty, and point to this as a key area of research to be able to constrain these estimates further.

‘We recalculate total krill FP_{100} fluxes based on maximum ($6.29 \text{ mg C ind.}^{-1} \text{ d}^{-1}$) and minimum ($0.67 \text{ mg C ind.}^{-1} \text{ d}^{-1}$) literature estimates (Belcher et al., 2017). This results in a range in seasonally integrated FP production ($FP_{100,SEA}$) of $0.008\text{--}0.079 \text{ GT C yr}^{-1}$ (Table 2) highlighting the need for further studies on krill FPP rates to constrain this parameter or accurately model its variability (perhaps based on factors such as food availability).’

References

Atkinson, A., Siegel, V., Pakhomov, E. A. and Rothery, P.: Long-term decline in krill stock and increase in salps within the Southern Ocean, *Nature*, 432(November), 100–103, doi:10.1038/nature02950.1., 2004.

Atkinson, A., Shreeve, R. S., Hirst, A. G., Rothery, P., Tarling, G. A., Pond, D. W., Korb, R. E., Murphy, E. J. and Watkins, J. L.: Natural growth rates in Antarctic krill (*Euphausia superba*): II . Predictive models based on food , temperature , body length , sex , and maturity stage, *Limnol. Oceanogr.*, 51(2), 973–987, doi:10.4319/lo.2006.51.2.0973, 2006.

Atkinson, A., Siegel, V., Pakhomov, E. A., Rothery, P., Loeb, V., Ross, R. M., Quetin, L. B., Schmidt, K., Fretwell, P., Murphy, E. J., Tarling, G. A. and Fleming, A. H.: Oceanic circumpolar habitats of Antarctic krill, *Mar. Ecol. Prog. Ser.*, 362, 1–23, doi:10.3354/meps07498, 2008.

Atkinson, A., Siegel, V., Pakhomov, E. A., Jessopp, M. J. and Loeb, V.: A re-appraisal of the total biomass and annual production of Antarctic krill, *Deep. Res. Part I Oceanogr. Res. Pap.*, 56(5), 727–740, doi:10.1016/j.dsr.2008.12.007, 2009.

Atkinson, A., Schmidt, K., Fielding, S., Kawaguchi, S. and Geissler, P. A.: Variable food absorption by Antarctic krill: Relationships between diet, egestion rate and the composition and sinking rates of their fecal pellets, *Deep Sea Res. Part II Top. Stud. Oceanogr.*, 59–60, 147–158, doi:10.1016/j.dsr2.2011.06.008, 2012.

Atkinson, A., Hill, S. L., Pakhomov, E. A., Siegel, V., Anadon, R., Chiba, S., Daly, K. L., Downie, R., Fielding, S., Fretwell, P., Gerrish, L., Hosie, G. W., Jessopp, M. J., Kawaguchi, S., Krafft, B. A., Loeb, V., Nishikawa, J., Peat, H. J., Reiss, C. S., Ross, R. M., Quetin, L. B., Schmidt, K., Steinberg, D. K., Subramaniam, R. C., Tarling, G. A. and Ward, P.: KRILLBASE: A circumpolar database of Antarctic krill

and salp numerical densities, 1926–2016, *Earth Syst. Sci. Data*, 9(1), 193–210, doi:10.5194/essd-9-193-2017, 2017.

Belcher, A., Tarling, G. A., Manno, C., Atkinson, A., Ward, P., Skaret, G., Fielding, S., Henson, S. A. and Sanders, R.: The potential role of Antarctic krill faecal pellets in efficient carbon export at the marginal ice zone of the South Orkney Islands in spring, *Polar Biol.*, 40(10), doi:10.1007/s00300-017-2118-z, 2017.

Cadée, G. C.: Organic carbon and its sedimentation during the ice retreat period in the Wedell-Scotia Sea, 1988, *Polar Biol.*, 12, 253–259, doi:10.1007/BF00238267, 1992.

Cadée, G. C., González, H. E. and Schnack-Schiel, S. B.: Krill diet affects faecal string settling, *Polar Biol.*, 12(1), 75–80, doi:10.1007/BF00239967, 1992.

Clarke, A., Quetin, L. B. and Ross, R. M.: Laboratory and field estimates of the rate of faecal pellet production by Antarctic krill, *Euphausia superba*, *Mar. Biol.*, 98(4), 557–563, doi:10.1007/BF00391547, 1988.

Cox, M. J., Candy, S., Mare, W. K. De, Nicol, S., Kawaguchi, S. and Gales, N.: No evidence for a decline in the density of Antarctic krill *Euphausia superba* Dana, 1850, in the Southwest Atlantic sector between 1976 and 2016, *J. Crustac. Biol.*, 1–6, doi:10.1093/jcabi/ruy072, 2018.

Dilling, L. and Alldredge, A. L.: Fragmentation of marine snow by swimming macrozooplankton: A new process impacting carbon cycling in the sea, *Deep. Res. Part I Oceanogr. Res. Pap.*, 47(7), 1227–1245, doi:10.1016/S0967-0637(99)00105-3, 2000.

Dunne, J. P., Armstrong, R. A., Gnanadesikan, A. and Sarmiento, J. L.: Empirical and mechanistic models for the particle export ratio, *Global Biogeochem. Cycles*, 19(4), doi:10.1029/2004GB002390, 2005.

Everson, I. and Bone, D. G.: The effectiveness of the RMT8 system for sampling krill (*Euphausia superba*) swarms, *Polar Biol.*, 6, 83–90, doi:10.1007/BF00258257, 1986.

Fielding, S., Watkins, J. L., Collins, M. A., Enderlein, P. and Venables, H. J.: Acoustic determination of the distribution of fish and krill across the Scotia Sea in spring 2006, summer 2008 and autumn 2009, *Deep. Res. Part II*, 59–60, 173–188, doi:10.1016/j.dsr2.2011.08.002, 2012.

González, H. E.: The distribution and abundance of krill faecal material and oval pellets in the Scotia and Weddell Seas (Antarctica) and their role in particle flux, *Polar Biol.*, 12, 81–91, doi:10.1007/BF00239968, 1992.

Henson, S. A., Sanders, R., Madsen, E., Morris, P. J., Le Moigne, F. and Quartly, G. D.: A reduced estimate of the strength of the ocean's biological carbon pump, *Geophys. Res. Lett.*, 38(4), L04606, doi:10.1029/2011GL046735, 2011.

Hewitt, R. P., Watkins, J., Naganobu, M., Sushin, V., Brierley, A. S., Demer, D., Kasatkina, S., Takao, Y., Goss, C., Malyshko, A., Brandon, M., Kawaguchi, S., Siegel, V., Trathan, P., Emery, J., Everson, I. and Miller, D.: Biomass of Antarctic krill in the Scotia Sea in January/February 2000 and its use in revising an estimate of precautionary yield, *Deep Sea Res. Part II Top. Stud. Oceanogr.*, 51, 1215–1236, doi:10.1016/j.dsr2.2004.06.011, 2004.

Iversen, M. H., Pakhomov, E. A., Hunt, B. P. V., Jagt, H. Van Der, Wolf-gladrow, D. and Klaas, C.: Sinkers or floaters? Contribution from salp pellets to the export flux during a large bloom event in the Southern Ocean, *Deep Sea Res. Part II Top. Stud. Oceanogr.*, 138(December 2016), 116–125, doi:10.1016/j.dsr2.2016.12.004, 2017.

Martin, B., Koppelman, R. and Kassatov, P.: Ecological relevance of salps and doliolids in the

northern Benguela Upwelling System, *J. Plankton Res.*, 39, 290–304, doi:10.1093/plankt/fbw095, 2017.

Martin, J. H., Knauer, G. A., Karl, D. M. and Broenkow, W. W.: VERTEX: carbon cycling in the northeast Pacific, *Deep Sea Res. Part I Oceanogr. Res. Pap.*, 34(2), 267–285, doi:10.1016/0198-0149(87)90086-0, 1987.

Meyer, B., Freier, U., Grimm, V., Groeneveld, J. and Hunt, B. P. V: The winter pack-ice zone provides a sheltered but food-poor habitat for larval Antarctic krill, *Nat. Ecol. Evol.*, 1(December), 1853–1861, doi:10.1038/s41559-017-0368-3, 2017.

Poulsen, L. and Iversen, M. H.: Degradation of copepod fecal pellets: key role of protozooplankton, *Mar. Ecol. Prog. Ser.*, 367, 1–13, doi:10.3354/meps07611, 2008.

Poulsen, L. and Kiørboe, T.: Coprophagy and coprorhexy in the copepods *Acartia tonsa* and *Temora longicornis*: clearance rates and feeding behaviour, *Mar. Ecol. Prog. Ser.*, 299, 217–227, doi:10.3354/meps299217, 2005.

Ramaswamy, V., Sarin, M. M. and Rengarajan, R.: Enhanced export of carbon by salps during the northeast monsoon period in the northern Arabian Sea, *Deep Sea Res. Part II Top. Stud. Oceanogr.*, 52, 1922–1929, doi:10.1016/j.dsr2.2005.05.005, 2005.

Smith, Jr., K. L., Sherman, A. D., Huffard, C. L., McGill, P. R., Henthorn, R., Von Thun, S., Ruhl, H. A., Kahru, M. and Ohman, M. D.: Large salp bloom export from the upper ocean and benthic community response in the abyssal northeast Pacific: Day to week resolution, *Limnol. Oceanogr.*, 59(3), 745–757, doi:10.4319/lo.2014.59.3.0745, 2014.

Stone, J. P. and Steinberg, D. K.: Long-term time-series study of salp population dynamics in the Sargasso Sea, *Mar. Ecol. Prog. Ser.*, 510, 111–127, doi:doi.org/10.3354/meps10985, 2014.

Stone, J. P. and Steinberg, D. K.: Salp contributions to vertical carbon flux in the Sargasso Sea, *Deep. Res. Part I Oceanogr. Res. Pap.*, 113, 90–100, doi:10.1016/j.dsr.2016.04.007, 2016.

Wexels-Riser, C., Wassmann, P., Olli, K. and Arashkevich, E.: Production, retention and export of zooplankton faecal pellets on and off the Iberian shelf, north-west Spain, *Prog. Oceanogr.*, 51(2–4), 423–441, doi:10.1016/S0079-6611(01)00078-7, 2001.

Wexels Riser, C., Reigstad, M., Wassmann, P., Arashkevich, E. and Falk-Petersen, S.: Export or retention? Copepod abundance, faecal pellet production and vertical flux in the marginal ice zone through snap shots from the northern Barents Sea, *Polar Biol.*, 30, 719–730, doi:10.1007/s00300-006-0229-z, 2007.

Reviewers' comments:

Reviewer #1 (Remarks to the Author):

Major comments:

I believe the authors have gone some way to address my concerns, but not fully. On line 14, I agree with the use of the word "underrepresented", but I disagree with the last sentence of the abstract. I don't think the authors can claim that the impact of FP export is not captured by many global biogeochemical models. As I explained in my previous review, most global biogeochemical models are tuned using the climatological database of nutrients, oxygen, and carbon. Therefore, to the extent that FP export has a significant impact on nutrient and carbon distributions, their mean impact will be captured. Only if FP fluxes have a minimal impact on nutrients and oxygen concentrations will they be unaccounted for. Furthermore, I don't buy the argument on line 192 that the spatio-temporal resolution of the WOA is inadequate to capture the impact of episodic fluxes of krill FP. The flux events are short lived, but their impact gets integrated by the tracer fields and are much longer lived and of much larger spatial extent.

A major strength of inverse models is that they are insensitive to the details of the underlying small-scale processes. This is in sharp contrast to bottom-up models that sum up individual processes. If for some reason we were to learn that krill abundances are twice as high as previously thought, then we would want to revise our estimates of carbon export based on bottom-up models, but our estimates based on top-down inverse models would remain unchanged. In terms of interpreting the current biogeochemical state of the ocean, global biogeochemical models behave as inverse models because they are tuned against climatological tracer data. It is for the purpose of predicting future changes that models need to get the detailed processes right, and I think it is for this problem that the present work is most important. In my opinion, the significance of the present work is that it shows that krill FP export fluxes are important contributors to the global carbon export and therefore need to be properly represented in mechanistic models that aim at predicting future changes. The claim that the present work implies that there is a need to revise current carbon export estimates is not supported by any evidence.

Minor comments:

line 114 delete the word "flux" to be consistent with the units given in GT C without a per area and/or per time.

line 127 delete the word "flux". Otherwise it sounds as if you are comparing a flux to a total export.

line 174 the word "flux" can also be deleted to make the sentence more clear.

Reviewer #2 (Remarks to the Author):

The authors have adequately addressed the concerns of the reviewers and revised the manuscript accordingly.

Reviewer #1 (Remarks to the Author):

Major comments:

I believe the authors have gone some way to addressing my concerns, but not fully. On line 14, I agree with the use of the word "underrepresented", but I disagree with the last sentence of the abstract. I don't think the authors can claim that the impact of FP export is not captured by many global biogeochemical models. As I explained in my previous review, most global biogeochemical models are tuned using the climatological database of nutrients, oxygen, and carbon. Therefore, to the extent that FP export has a significant impact on nutrient and carbon distributions, their mean impact will be captured. Only if FP fluxes have a minimal impact on nutrients and oxygen concentrations will they be unaccounted for. Furthermore, I don't buy the argument on line 192 that the spatio-temporal resolution of the WOA is inadequate to capture the impact of episodic fluxes of krill FP. The flux events are short lived, but their impact gets integrated by the tracer fields and are much longer lived and of much larger spatial extent.

A major strength of inverse models is that they are insensitive to the details of the underlying small-scale processes. This is in sharp contrast to bottom-up models that sum up individual processes. If for some reason we were to learn that krill abundances are twice as high as previously thought, then we would want to revise our estimates of carbon export based on bottom-up models, but our estimates based on top-down inverse models would remain unchanged. In terms of interpreting the current biogeochemical state of the ocean, global biogeochemical models behave as inverse models because they are tuned against climatological tracer data. It is for the purpose of predicting future changes that models need to get the detailed processes right, and I think it is for this problem that the present work is most important. In my opinion, the significance of the present work is that it shows that krill FP export fluxes are important contributors to the global carbon export and therefore need to be properly represented in mechanistic models that aim at predicting future changes. The claim that the present work implies that there is a need to revise current carbon export estimates is not supported by any evidence.

Response to reviewers

REPLY: We thank the reviewer for adding clarification to the point that we did not properly pick up in the previous round. The reviewer makes the point that the strength of our paper is in revealing the important role of krill in POC export, and we have thus strengthened the manuscript around this point, ensuring that we are clear about the benefit of our work for improving the predictive capabilities of models under future scenarios.

Reviewer #1 suggests the rephrasing of two specific statements in our revised manuscript. We agree that the mean impact of krill FP fluxes should be captured in changes in the nutrient fields due to the longer integration time. We state in the manuscript (methods lines 390-391) that the biogeochemical models we compare to (Primeau et al. (2013), MacCreedy et al. 2001) and Schlitzer et al. (2002) provide better spatial coverage of the MIZ than satellite-based measurements as they are not reliant on snapshot style POC measurements. Additionally, we agree that our work highlights the significant contribution that krill FP make

to export flux. In line with the recommendations of the reviewer, we suggest in the manuscript that their incorporation in mechanistic models could make improvements to POC export estimates for the Southern Ocean that justify the increased model complexity.

We have been through the manuscript and made a number of revisions to clarify our arguments based on the reviewer's comments. We have rephrased the specific statements highlighted by Reviewer #1 and in particular, we have clarified that proposed model improvements are via mechanistic representation of the flux of krill FP. We refer to line numbers in the unmarked revised manuscript that we have submitted with this response to reviewers.

We have amended the following:

Abstract, lines 20-22:

“The magnitude of our conservatively estimated flux highlights the important role of large, swarming macrozooplankton in POC export and the need to incorporate such processes more mechanistically to improve model projections.”

Introduction, lines 55-68 in the previous revised version, has been edited, to remove the statements about the spatial and temporal coverage of WOA data used in biogeochemical models that are tuned to nutrient fields. We restrict our discussion to empirical models utilising sparse POC flux measurements, and highlight the importance of including krill mechanistically to able accurate future projections of POC export. The paragraph (lines 50-59) now reads:

“Measuring the flux of POC in situ at various depths in the water column is challenging, resulting in relatively poor spatial and temporal coverage of measurements when compared to more easily measureable ocean variables such as temperature or nitrate. Empirically derived models for POC flux that utilise limited in situ ‘snapshot’ POC flux measurements may not capture episodic fluxes, such as those driven by krill swarms, and likely underestimate the carbon flux in regions of high krill density. This is especially true for models with a spatially and temporally invariant attenuation rate (typically Martin’s b value¹⁴). This means that individual krill swarms will not be resolved either in time or space. Without mechanistic representation of the FP flux associated with krill swarms, biogeochemical models may not make accurate projections of the contribution of the Southern Ocean to global carbon export.”

In line with the reviewer's suggestions, we have also revised lines 84-86, to make clear the strength of our work in highlighting the contribution that krill FP can make to the POC flux, and their lack of mechanistic inclusion in models:

“These results highlight that krill FPs are an important contributor to carbon flux and yet they are not mechanistically represented in global biogeochemical models, which restricts our ability to predict future changes to the BCP.”

With reference to the reviewer's comment of ‘the argument on line 192 that the spatio-temporal resolution of the WOA is inadequate’, we have removed this statement, and the paragraph in the discussion (lines 177-183) now reads:

“We estimate that krill FP fluxes represent 12.5-17.7 % of total carbon export (Table 1) based on the models of MacCreedy et al. (2001) and Primeau et al. (2013). Krill FP fluxes could therefore represent a significant fraction of Southern Ocean POC export flux and their mechanistic inclusion in global models could improve projections of future ocean carbon

uptake. The relatively sparse data availability of POC flux measurements in the Southern Ocean, and the lack of the mechanistic inclusion of krill FP export may lead to underestimations of the contribution of the Southern Ocean to global export fluxes.”

We have added the following sentence (lines 268-271) to the closing paragraph of the discussion:

“Additionally, our work highlights that krill FP fluxes need to be mechanistically represented in global biogeochemical models to enable more accurate projections of the future Southern Ocean contribution to carbon export.”

Minor comments:

line 114 delete the word "flux" to be consistent with the units given in GT C without a per area and/or per time.

REPLY: Deleted

line 127 delete the word "flux". Otherwise it sounds as if you are comparing a flux to a total export.

REPLY: Deleted

line 174 the word "flux" can also be deleted to make the sentence more clear.

REPLY: Deleted

Reviewer #2 (Remarks to the Author):

The authors have adequately addressed the concerns of the reviewers and revised the manuscript accordingly.

REPLY: Thank you for taking the time to review our manuscript again.

REVIEWERS' COMMENTS:

Reviewer #1 (Remarks to the Author):

I believe that the authors have addressed all the issues raised in my previous reviews and I would be delighted to see this work published in Nature Communications.